# Adverse Childhood Experiences and Prescription Stimulant Use in Adults: A Systematic Review

**DOI:** 10.3390/pharmacy12020052

**Published:** 2024-03-21

**Authors:** Avery Beatty, Emily Shepard, Erek Bickford, Lisa Weyandt

**Affiliations:** 1Department of Psychology, University of Rhode Island, Kingston, RI 02881, USA; 2Interdisciplinary Neuroscience Program, University of Rhode Island, Kingston, RI 02881, USA; 3Department of Psychology and Interdisciplinary Neuroscience Program, University of Rhode Island, Kingston, RI 02881, USA

**Keywords:** adverse childhood experiences, prescription stimulant misuse, substance use, trauma

## Abstract

Nonmedical prescription stimulant use (NPSU) is an ongoing public health crisis in the United States. There exists, however, a dearth of research investigating specific childhood risk factors that may contribute to this illicit use. Adverse childhood experiences (ACEs) may be a significant risk factor to address in the prevention and treatment of NPSU, as research has provided evidence for increased substance use, potentially including prescription stimulant misuse, among individuals who have experienced trauma. A systematic review of the literature was conducted to gather and synthesize research articles specifically examining the relationship between nonmedical prescription stimulant use in adults and the experience of adverse childhood experiences (ACEs). Four studies met all inclusion criteria and were included in the review. The results demonstrated that the nonmedical use of prescription stimulants in adults is significantly associated with ACEs, with a higher numerical count of ACEs associated with a greater likelihood of nonmedical prescription stimulant use. Gaps in the literature were identified, specifically noting a lack of information available regarding the relationship between ACEs and NPSU within gender- and racially diverse populations. The findings have implications for informing interventions related to ACEs and the misuse of prescription stimulant medication in adults.

## 1. Introduction

### 1.1. Nonmedical Prescription Stimulant Use

Prescription stimulant medications, such as dextroamphetamine (e.g., Adderall) and lisdexamfetamine dimesylate (e.g., Vyvanse), are a commonly used and effective treatment for children and adults with attention-deficit/hyperactivity disorder (ADHD) when used as prescribed [1,2,3]. There are common dose-dependent side effects associated with stimulant medications, such as decreased appetite, weight loss, dehydration, difficulty sleeping, and anxiety [1,2,3]. When prescription stimulant medications are misused (i.e., illicitly used without a valid prescription or for reasons other than prescribed), more severe side effects have been observed. These include risky sexual behaviors, psychosis, suicidal ideation, and reliance on medication [4,5]. Prescription stimulants have a potential for psychological and physiological dependence and are classified by the Drug Enforcement Administration (DEA) as Schedule II medications [6]. Despite the potential health and legal consequences, nonmedical prescription stimulant use (NPSU) remains an ongoing public health concern in the United States [5,7,8].

NPSU is also particularly problematic across college campuses, with reported prevalence rates ranging from 14% to 34% among college students in the United States [7,9,10]. Previous research has demonstrated that, as an increasing number of individuals with ADHD attend college, the availability and illicit use of prescription stimulant medications among students without ADHD has also increased [11,12,13]. In addition to easy access to nonmedical prescription stimulants, college students experience unique social and academic stressors that often precipitate NPSU. For example, research has shown that motivations for NPSU among college students include both academic (e.g., increased concentration and focus and improved test performance) and recreational purposes (i.e., partying) [7,9,10]. Additionally, NPSU seems to persist post-graduation. Holt and McCarthy (2020) found that among recent college graduates with a history of NPSU, approximately 50% continued to engage in NPSU after graduating [14].

Given the increasing rates of NPSU across the United States, as well as the harmful effects associated with this illicit use, it is crucial to identify the potential risk factors that can inform the development of effective prevention and intervention methods. Research has increasingly focused on examining how childhood factors influence the trajectories of substance use in adulthood. However, there is limited research specifically exploring the childhood risk factors that may contribute to NPSU. Adverse childhood experiences may be a significant risk factor to consider. Research has substantiated that adverse childhood experiences are linked to increased substance use, generally [15,16,17]. A study involving the consolidation of research that specifically explores the relationship between adverse childhood experiences and NPSU does not exist, however.

### 1.2. Adverse Childhood Experiences

Adverse childhood experiences (ACEs) refer to potentially traumatic events that occur before the age of 18 and can have detrimental effects on an individual’s health and well-being across the lifespan [15,17,18,19]. Felitti and colleagues (1998) first operationalized ACEs as exposure to abuse (e.g., sexual, physical, and emotional) and household dysfunction (e.g., substance use by a household member, household member with a mental illness, witnessing domestic violence, and familial incarceration). Other studies have also included additional ACEs, such as neglect (e.g., emotional and physical), parental separation, loss of family members and/or friends, longstanding financial hardship, and community violence [15,17,18,19]. ACEs affect a large proportion of the adult population, with approximately 62% of adults reporting an experience of at least one ACE [6,20]. It is well substantiated in the literature that ACEs are associated with detrimental health outcomes and adverse health risk behaviors [15,17,18,19]. Physiological research has substantiated that exposure to ACEs can lead to harmful levels of stress and the persistent activation of the stress response system. This stress response can alter the nervous, endocrine, and immune systems, which adversely affects an individual’s executive functioning, decision-making abilities, impulse control, brain reward systems, and emotion regulation throughout the lifespan [3,15,16]. Similarly, previous work has found that ACEs are significantly associated with emotion regulation difficulties in adulthood (i.e., an individual’s ability to effectively manage distressing emotions). In line with the self-medication hypothesis, adults with a history of ACEs may turn to illicit substances such as nonmedical prescription stimulants to stimulate their mood and cope with their emotions (e.g., [21,22]).

Extant research has demonstrated a strong relationship between ACEs and illicit substance use [15,17,18,19]. In fact, ACEs are among the most robust predictors of substance use in adulthood, and research has consistently found that ACEs are significantly associated with the initiation of illicit substance use at an early age [15,17,18,19]. Additionally, recent studies have also shown a cumulative effect between ACEs and substance use. Specifically, as the number of ACE exposures increases, the risk of developing a substance use disorder also increases [19,23]. Greater instances of ACEs were also found to be associated with an increased risk for more severe substance use, including the earlier initiation of more dangerous illicit substances (e.g., injection of drugs) and increased rates of overdoses [16,17,24]. Despite the well-established associations between ACEs and substance use in adulthood, critical questions about the relationship between ACEs and the misuse of prescription stimulants in adults still exist.

The purpose of the present paper is to systematically review and summarize the literature within the past ten years regarding the association between NPSU and ACEs among young adult and adult populations. The present review aims to (a) determine the percentage of studies on NPSU that included an investigation of ACES and/or traumatic childhood events; (b) explore the specific association between ACEs and NPSU among young adult and adult populations; (c) identify trends of ACEs experienced by young adults and adults who misuse prescription stimulant medications; (d) differentiate between specific prescription stimulant medications commonly misused by young adults and adults with a history of ACE(s); and (e) determine the prevalence of demographic factors such as age, gender, and race and/or ethnicity that may be linked to ACEs and prescription stimulant use in young adults and adults.

## 2. Methods

### 2.1. Search and Retrieval

Following the Preferred Reporting Items for Systematic Reviews and Meta-Analyses (PRISMA) guidelines, a systematic review of the literature was conducted to examine the association between NPSU and ACEs among adults [25]. PRISMA is an evidence-based framework and is designed to facilitate a comprehensive search of databases using specific keywords and search terms, screening articles following a review of the title, abstract, and full text [25]. Consistent with previous research, NPSU was operationalized as the use of prescription stimulants without a valid prescription or for reasons other than prescribed (e.g., more frequent use, using higher doses, combining with other illicit substances, and/or for nonmedical purposes). Furthermore, ACEs were operationalized as any traumatic event that occurred before the age of 18, encompassing abuse (e.g., sexual, physical, or emotional), neglect (e.g., physical or emotional), and/or household dysfunction (e.g., a household member with mental illness, witnessing domestic violence, divorce, familial incarceration, etc.) [23].

All researchers were trained in the systematic review study protocol before article and data retrieval. A combination of the following keywords were searched within the bibliographic databases PubMed and PsycINFO: nonmedical prescription stimulant use AND Adverse Childhood Experiences (ACEs); nonmedical prescription stimulant use AND early trauma; prescription stimulant use AND Adverse Childhood Experiences (ACEs); nonmedical prescription stimulant use AND Adverse Childhood Experiences (ACEs) AND trauma AND Methylphenidate OR Amphetamines/Dextroamphetamine OR Lisdexamfetamine OR Vyvanse OR Adderall OR Ritalin OR Concerta. The search included articles that were published between 1 January 2013 and 1 May 2023.

### 2.2. Eligibility Criteria

#### 2.2.1. Inclusion Criteria

The study investigated the relationship between nonmedical prescription stimulant use and ACEs;The study included only adult participants;The sample size of the study was n > 1;The study was published in the last 10 years;The study was originally published in English;The study used an original or secondary dataset;The study was published in a peer-reviewed journal.

#### 2.2.2. Exclusion Criteria

The study did not explicitly identify nonmedical prescription stimulant misuse;The study did not explicitly examine the relationship between ACEs and nonmedical prescription stimulant use;The study did not include ACEs and prescription stimulants use, or misuse was investigated in nonadult samples (i.e., child and adolescent samples);Meta-analyses, systematic reviews, protocols for future research, and case studies were excluded;The study was not published in English;The study was not published in a peer-reviewed journal.

#### 2.2.3. Data Extraction

The studies included in this review were evaluated, and information relevant to study goals was extracted and compiled into one table (i.e., see Table 1). The following information was extracted from each article: sample size and description, specific ACEs and prescription stimulant medications studied, and sample demographics (i.e., age, gender, race, and ethnicity).

## 3. Results

### 3.1. Study Selection

A total of 398 articles were identified via the bibliographic databases PubMed and PsycINFO. After systematic reviews, meta-analyses, and articles not published in English were excluded, the remaining (N = 165) articles were screened based on title and abstract review. The remaining articles (N = 36) then underwent a full-text review. The full-text review resulted in four final studies that met the eligibility criteria and were included in the systematic review; see Figure 1 for more details.

### 3.2. Summary of Included Articles

#### 3.2.1. Review of NPSU and ACEs

Regarding the first research aim of identifying the percentage of studies on NPSU that also examined ACEs, our findings revealed that only four studies in the last 10 years specifically investigated the relationship between nonmedical prescription stimulant use in adulthood and ACEs, representing approximately 1.01% of all the studies yielded by the search criteria. The sample sizes of the four studies ranged from 219 to 36,309. Combined, the studies yielded 46,579 total participants.

#### 3.2.2. Adverse Childhood Experiences and NPSU

Regarding the aim to explore the specific associations between ACEs and NPSU among adults, our findings revealed that all four studies found a significant, positive relationship between ACEs and NPSU in adulthood. All four studies also found that greater ACEs were associated with increased stimulant use. In one study (e.g., Tang et al., 2021), the authors investigated whether ACEs were associated with the earlier initiation of nonmedical prescription stimulants, and their findings revealed that participants with earlier age of first use also reported a greater number of ACEs [23]. Additionally, only one study noted specific relationships between race and ethnicity, ACEs, and NPSU. Specifically, Pakdaman and colleagues (2018) found that Asian Pacific Islanders (APIs) and Hispanics who experienced ACEs were significantly more likely to misuse prescription stimulants than White and Black participants with ACEs [19].

Regarding the specific aim of exploring how ACEs were operationalized in each study and the identification of any trends and/or patterns regarding specific ACEs and NPSU, the results revealed that ACEs were similarly defined in all four of the included studies. Each study included sexual, physical, and emotional abuse in their operationalization of ACEs. There were common trends observed across some of the studies. For example, Pakdaman et al. (2022) and Fairman et al. (2021) included parental substance use and parental mental health in their operationalization of ACEs, while Tang et al. (2020) and Forster et al. (2018) extended this definition to include household substance use and mental illness, as well as witnessing and/or experiencing domestic violence [19,23,24,26]. Tang et al. (2020) also included household incarceration and parental separation or divorce [23]. Finally, Pakdaman et al. (2022) was the only study that specifically included housing insecurity as an ACE [19]. In line with the second aim of exploring the general relationship between ACEs and NPSU, the findings from each study similarly revealed that each form of abuse (e.g., sexual, physical, and emotional), as well as each cluster of adverse events (e.g., household dysfunction), was significantly associated with NPSU. Regarding the third aim of identifying specific ACEs associated with NPSU, across the four studies, childhood emotional abuse, divorce, and parental substance use were the most common ACEs associated with NPSU.

#### 3.2.3. Specific Prescription Stimulant Medications and ACEs

Furthermore, regarding the fourth research aim, which aimed to differentiate between specific prescription stimulant medications commonly misused by adults with a history of ACEs, our findings revealed that none of the included studies delineated between different prescription stimulant medications and ACEs. From the narrowest perspective, Pakdaman and colleagues (2022) investigated the relationship specifically between only Adderall and ACEs, with no other prescription stimulant medications included in their analyses [19]. Forster et al. (2018), on the other hand, studied the relationship between the nonmedical use of Ritalin and/or Adderall with ACEs [24]. No distinction was made between the two medications, and other stimulant medications were not included for analysis. Similarly, Fairman et al. (2021) combined the misuse of Adderall, Vyvanse, Ritalin, and Concerta into one prescription stimulant variable and studied the relationship between the misuse of any of these medications with ACEs [26]. Even more broadly, Tang et al. (2020) did not specifically delineate the independent associations of prescription stimulant medications and ACEs, instead comparing the misuse of any prescription stimulant medication with no history of misuse [23].

#### 3.2.4. Summary of Demographic Findings: Age, Gender, Race, Ethnicity, Specific Populations

The final aim of this review was to describe the incorporation and reporting of demographic factors (e.g., age, gender, race, and ethnicity) across each study. Regarding age, differences were seen across studies; however, it was generally found that age was limited to young adulthood. Three studies investigated the relationship between nonmedical prescription stimulant use and ACEs among college student populations. Participants from these three studies were volunteers, and their data were derived from two national college health assessments (e.g., American College Health Association’s National College Health Assessment II; Project Documenting Experiences with Cigarettes and Other Tobacco in Youth). Moreover, Tang and colleagues (2018) investigated NPSU and ACEs among a nationally representative sample of the noninstitutionalized adult population 18 years old or older in the United States. Participants were asked to take part in the National Epidemiological Survey on Alcohol and Related Conditions (NESARC-III) conducted in 2012–2013 in person by the National Institute on Alcohol Abuse and Alcoholism (NIAAA). While Tang and colleagues (2021) omitted age apart from specifying that participants were all over the age of 18, the average participant age of the remaining three studies was approximately 22 years old. Furthermore, concerning gender, a majority of the studies (n = 3) reported having a predominantly female sample [19,24,26]. All three of these studies reported gender according to biological binary, with no inclusion of other gender identities. The fourth study did not include any data on the distribution of genders within the sample [23].

In line with the recent literature, only studies that reported the distribution of *at least* three different racial and/or ethnic groups were considered to have fully reported racial demographics, and studies that only reported one or two different racial and/or ethnic groups were considered to have partially reported racial demographics [27]. Two studies fully reported racial demographics [19,24]. Specifically, Pakdaman et al. (2021) noted that 55% of their sample identified as Hispanic, 27.9% as White, and 17% as Asian Pacific Islander [19]. Forster et al. (2018) reported that 37% of their sample identified as Hispanic, 33% as White, 15% as Asian Pacific Islander, 7% as Biracial, 4% as Black, and 4% as Other [24]. Moreover, the partial reporting of racial and/or ethnic group demographics was evident in the study by Fairman et al. (2021), in which they grouped participants by ethnicity, including Non-Hispanic White (82%) and Non-White (18%) [26]. Lastly, Tang et al. (2021) did not report any information regarding race or ethnicity [23].

## 4. Discussion

The purpose of the current paper was to systematically review and summarize the literature within the past ten years on the relationship between NPSU and ACEs among young adult and adult populations. Specifically, the present review sought to (a) determine the percentage of studies on nonmedical prescription stimulant use that included an investigation of adverse childhood experiences (ACEs) and/or traumatic childhood events; (b) explore the specific association between ACEs and nonmedical prescription stimulant use among young adult and adult populations; (c) identify trends of ACEs experienced by young adults and adults who misuse prescription stimulant medications; (d) differentiate between specific prescription stimulant medications commonly misused by young adults and adults with a history of ACE(s); and (e) determine the prevalence of demographic factors such as age, gender, race, and ethnicity that may be associated with an increased risk of nonmedical prescription stimulant use.

The results revealed that, of the 398 articles centering on nonmedical prescription stimulant use conducted within the past decade (2013–2023), only 1.01% (N = 4) specifically investigated ACEs as a risk factor for NPSU, with all studies finding a significant association between ACEs and NPSU. Furthermore, the studies also demonstrated that greater ACEs were associated with increased stimulant use, as well as the initiation of use at a younger age [19,23,24,26]. Given the prevalence of ACEs among adults (i.e., approximately 62% of adults reported an experience of at least one ACE), these findings highlight the importance of prioritizing ACEs in health initiatives and interventions that are aimed at mitigating prescription stimulant misuse, particularly among college students. Additionally, the findings also underscore the importance of implementing trauma-informed care (i.e., understanding a person’s life experiences to deliver effective medical care) across healthcare settings, including substance use treatment. According to the Substance Abuse and Mental Health Services Administration (SAMHSA), trauma-informed care, especially within the context of substance use, has the potential to increase patient engagement and adherence to treatment and may lead to better health outcomes [28].

Furthermore, a majority of the studies did not explore the relationship between individual ACEs with the illicit use of different prescription stimulant medications. Most of the studies combined various stimulant medications into one prescription stimulant variable, while one study only investigated ACEs with the nonmedical use of one specific stimulant (e.g., Adderall). This approach limits the generalizability of results, given that prior research has demonstrated both dose- and medication-dependent effects of different stimulant medications. To adequately inform intervention and prevention programs, future research should investigate whether certain stimulants are more commonly misused among adults with ACEs, as well as identify if ACEs have idiosyncratic effects with specific prescription stimulant medications. Similarly, the results also highlight that there is a lack of standardization in how ACEs are measured and operationalized. This lack of standardization may impart added complexity when exploring what specific components of an ACE are predictive of future NPSU. Additionally, many existing ACE questionnaires that are utilized in the literature lack the inclusion of other known adversities, such as poverty and housing insecurity. Not including these in a measurement of ACEs is problematic, as they may have a mediating or direct effect on NPSU. Future research is needed to develop standardized ACE assessment measures.

Demographic differences (e.g., gender, race, and age) were also an important consideration in the current study. The majority of the sample across articles comprised women [19,23,24,29]. The inclusion of a primarily female sample is concerning given the robust evidence in the literature suggesting that men are more likely to misuse prescription stimulant medication than women [24]. Another concern that arose among the studies included in this systematic review was how gender was investigated and reported on. Of the three studies that reported gender demographics (e.g., [19,24,26]), each operationalized gender as a binary distribution of male and female participants. This is a critical oversight, as it excludes nonbinary and genderqueer individuals from the investigation of ACEs and nonmedical prescription stimulant use. The exclusion of greater gender diversity is particularly concerning given the recent literature demonstrating that gender minority populations experience higher levels of ACEs and discrimination compared to their cisgender peers, as well as the increased rates of comorbid substance and psychiatric disorders [30]. As such, future studies would greatly benefit from the inclusion of gender-minority populations in the investigation of ACEs and nonmedical prescription stimulant use.

Overall, half of the included studies (n = 2) reported on three or more races (e.g., Forster et al., 2018; Pakdaman et al., 2021), while Fairman et al. (2021) investigated ACEs and stimulant use among a predominately White population [19,24,26]. Concerningly, race or ethnicity was not included in any capacity in one of the four articles examining the relationship between NPSU and ACEs in the last decade [23]. Given the disproportional rates of ACEs experienced by people of color, it is critical to include diverse populations when investigating the association of ACEs and NPSU. For example, Pakdaman and colleagues (2018), who reported full racial demographic data, found that Asian Pacific Islanders (APIs) and Hispanics who experienced ACEs were significantly more likely to misuse prescription stimulants than White and Black participants with ACEs [19]. Additionally, the results of this systematic review highlighted that ethnicity was not separated from race for the majority of the articles. Future research should aim to delineate race and ethnicity as distinct concepts in order to avoid within-group homogeneity.

Regarding age, three of the four included studies investigated ACEs and prescription stimulant use in college populations. Although these designs are reflective of the increased use of stimulants among younger adults, research has also shown that the use of illicit prescription medications in older adults is gradually increasing. Specifically, illicit prescription medication use is more prevalent in older Americans than in adults in any other country [30]. Thus, future studies should aim to recruit and include older adults (i.e., above 44 years of age) in their participant samples.

Furthermore, this systematic review is not without limitations. The study did not include non-English articles, which might exclude relevant studies and/or result in language bias. Future research should consider including non-English articles if appropriate translation resources are available. Another major limitation of the current review is its limited and relatively homogenous sample. Specifically, the sample across all four studies was primarily White, female, and college-aged [19,23,24,26]. This sample composition may indicate a lack of generalizability to other ages, genders, races, and ethnicities. This is particularly concerning, as minoritized communities are not only at greater risk for ACEs and illicit substance use, but they also often face significant barriers to substance treatment, including poverty, racism, and structural barriers (e.g., poverty, differential access to treatment, etc.) [31]. This highlights the need for future research to be diverse in age, gender, and race to better understand the phenomenon of ACEs and NPSU among adults from various backgrounds to inform targeted interventions.

## 5. Conclusions

The nonmedical use of prescription stimulant medications is an ongoing public health concern across the United States. Despite the detrimental effects associated with illicit prescription stimulant use, there is a dearth of research investigating the risk factors that perpetuate this use. The current systematic review demonstrated that ACEs were significantly associated with nonmedical prescription stimulant use among young adults (i.e., college students) and adult samples and that greater ACEs were associated with increased stimulant use and initiation of use at an earlier age. These findings provide strong support for the integration of ACEs in intervention and prevention efforts aimed at decreasing illicit prescription stimulant use. These findings also underscore the importance of appropriate ACE screening in psychiatric clinics, as well as the need for college students to have access to trauma-informed programs and resources on campus. Advancing comprehensive strategies to prevent ACEs and treating underlying trauma among those using stimulants holds great promise to reduce stimulant use and its health and social consequences in the United States.

## Figures and Tables

**Figure 1 pharmacy-12-00052-f001:**
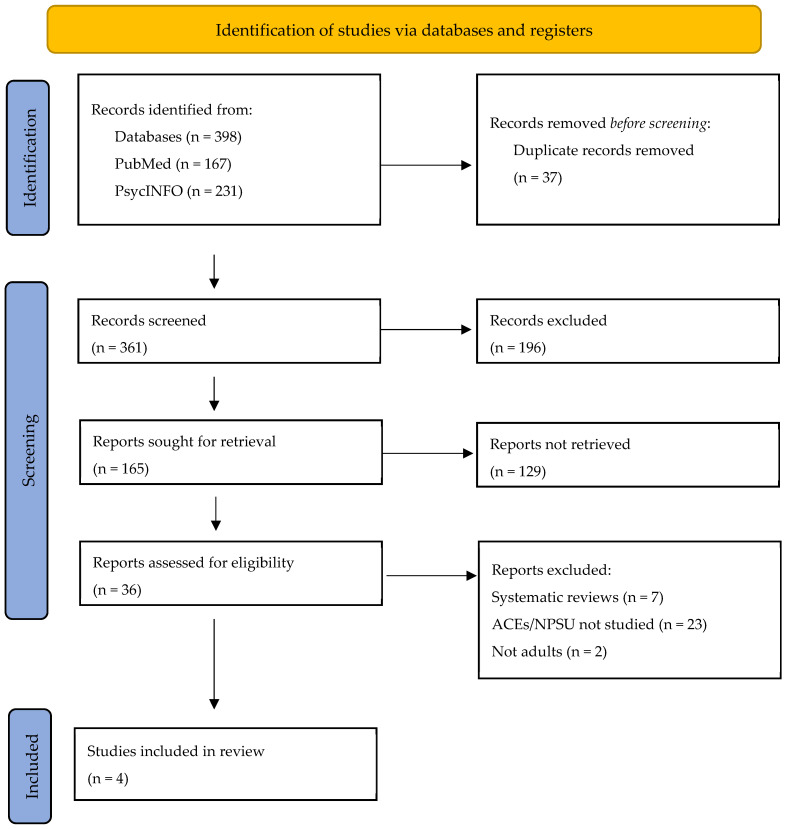
Flow diagram according to the PRISMA guidelines. This flow diagram was modeled after Page, M.J., McKenzie, J.E., Bossuyt, P.M., Boutron, I., Hoffmann, T.C., Mulrow, C.D., et al. The PRISMA 2020 statement: An updated guideline for reporting systematic reviews. *BMJ* 2021, 372, n71. https://doi.org/10.1136/bmj.n7 (accessed on 13 January 2024).

**Table 1 pharmacy-12-00052-t001:** Summary of Included Articles.

Article	Sample	ACEs	Stimulant(s)	Age (M)	Gender (Female)	Race and/or Ethnicity
[19]	3899 college students	Parental substance use & mental health; Verbal, physical, & sexual abuse; Homelessness	Adderall	24.55	69.9%	Hispanic: 55% White: 27.9% Asian/Pacific Islander: 16%
[23]	36,309 adults	Household: substance use, mental illness, incarceration, domestic violence, parental separation or divorce; Verbal, physical, emotional, & sexual abuse	Stimulant	N/A	N/A	N/A
[24]	2953 college students	Household substance use; Domestic violence; Verbal, physical, & sexual abuse	Ritalin, Adderall	21.34	73%	Hispanic: 37% White: 33%; Black: 4% Asian/Pacific Islander: 15% Multiracial 7%; Other: 4%
[26]	3418 college students	Parental mental health & substance use; Sexual, physical, & emotional abuse; Maltreatment	Adderall, Vyvanse, Ritalin, Concerta	20.72	54.8%	White: 82% Nonwhite: 18%

## Data Availability

Contributions presented in the study are included in the article, further inquiries can be directed to the corresponding authors.

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
