# Peer review of "Adverse Childhood Experiences and Prescription Stimulant Use in Adults: A Systematic Review"

_pharmacy, 2024, doi:10.3390/pharmacy12020052_

Round 1

Reviewer 1 Report

Comments and Suggestions for Authors

This systematic review found 4 papers examining how childhood adverse events might lead to prescription drug abuse later in life. I have a few comments that may improve the manuscript.

Introduction.

Line 2 you state that prescription stimulants are ‘safe’. I’d suggest that this has not yet been shown and, on the contrary, they may lead to further drug use. Please don’t say that they are safe.

Line 54. This sentence is confusing. Do you really mean that “….50% of young adults continued to misuse prescription stimulants after graduating from college.”

References 12-14 are mentioned in the introduction, how is this study different from these previous stuides? Please explicitly state what is new.

Methods.

Why did you just examine studies from the last 10 years. I suspect that there are numerous good studies from the previous 20-30 years. Increasing your data from 4 papers to more would be nice especially as 3 out of 4 of your papers are from college students. It would be good to get a more diverse adult population.

Results.

It would be nice to know what percentage of adults, who have had ACE, go on to prescription stimulant abuse, I don’t think this is mentioned anywhere?

It would be nice to know how these 4 studies found their participants. Were they all volunteers? Were they selected in some way or randomly approached?

Discussion and Conclusions.

I’d be very careful about your use of the word adult, as 3 of your 4 studies were from college populations. This population is hardly representative of all adults or of all ethnic backgrounds. I think you need a paragraph highlighting this limitation. You mention a “variety of adult populations” in your conclusions – I don’t think that you can say this.

Also, in the Conclusions you mention that increased ACE lead to more and earlier abuse of prescription stimulants – but where is this mentioned in the results? I didn’t see this anywhere else in the text.

Author Response

Introduction.

  • Line 2 you state that prescription stimulants are ‘safe’. I’d suggest that this has not yet been shown and, on the contrary, they may lead to further drug use. Please don’t say that they are safe.
    • Thank you for your comment, this has been addressed.

  • Line 54. This sentence is confusing. Do you really mean that “….50% of young adults continued to misuse prescription stimulants after graduating from college.”
    • This sentence has been clarified.

  • References 12-14 are mentioned in the introduction, how is this study different from these previous studies? Please explicitly state what is new.
    • This section has been edited, specifically stating that research has not been consolidated that examines the relationship between adverse childhood experiences and NPSU. specifically, and instead focuses only on general substance use.

Methods.

  • Why did you just examine studies from the last 10 years. I suspect that there are numerous good studies from the previous 20-30 years. Increasing your data from 4 papers to more would be nice especially as 3 out of 4 of your papers are from college students. It would be good to get a more diverse adult population.
    • We appreciate your comment. Extending the date span did not result in any further studies being included.

Results.

  • It would be nice to know what percentage of adults, who have had ACE, go on to prescription stimulant abuse, I don’t think this is mentioned anywhere?
    • Thank you for your comment. This was not one of the intended aims of the review.

  • It would be nice to know how these 4 studies found their participants. Were they all volunteers? Were they selected in some way or randomly approached?
    • Thank you for your comment. We included information about participant recruitment in the section titled Summary of Demographic Findings: Age, Gender, Race, Specific Populations.

Discussion and Conclusions.

  • I’d be very careful about your use of the word adult, as 3 of your 4 studies were from college populations. This population is hardly representative of all adults or of all ethnic backgrounds. I think you need a paragraph highlighting this limitation. You mention a “variety of adult populations” in your conclusions – I don’t think that you can say this.
    • Wording has been clarified throughout the paper to more accurately represent the sample. A paragraph has been added to address this and other limitations.

  • Also, in the Conclusions you mention that increased ACE lead to more and earlier abuse of prescription stimulants – but where is this mentioned in the results? I didn’t see this anywhere else in the text.
    • Thank you for your attention to detail. We clarified in the results section that ACEs lead to greater NPSU in all 4 studies, and that one study also found that greater ACEs were associated with earlier use.

Reviewer 2 Report

Comments and Suggestions for Authors

Thank you for allowing me to review this important review. I have a few questions/comments below.

Was this search registered with PROSPERO? If so, please include that information.

The search was about a year ago,  consider conducting another search to have articles published over the past year. As you noted, stimulant use research has been increasing.

Results:

As you noted that other genders we not included and the studies reported binary sex. Please also be sure to note it clearly that ethnicity was not separated from race for the majority of the articles.

Did the articles note any relationships between sex, gender, race, or ethnicity in the articles? You note the sample but not the analysis or findings of the articles. You make note in the discussion but not in the results.

Thank you

Author Response

  • The search was about a year ago,  consider conducting another search to have articles published over the past year. As you noted, stimulant use research has been increasing.
    • We did not find any other articles published since 2023.

Results:

As you noted that other genders we not included and the studies reported binary sex. Please also be sure to note it clearly that ethnicity was not separated from race for the majority of the articles.

  • Thank you for this comment. We included a discussion on this.

Did the articles note any relationships between sex, gender, race, or ethnicity in the articles? You note the sample but not the analysis or findings of the articles. You make note in the discussion but not in the results.

  • Thank you for this comment. We included a description of which studies noted specific relationships.

Reviewer 3 Report

Comments and Suggestions for Authors

Thank you for the opportunity to review this paper. The paper is a systematic review of ACEs and prescription stimulant use in adults. This paper is of clear importance given the public health crisis of nonmedical prescription stimulant use, yet, as the authors evidence, there is very limited research into the factors contributing to this. The paper is well structured and well written overall with minor grammatical errors to fix. I have highlighted a few additional comments which I think would be beneficial to address prior to publication.

Introduction

Line 44 – missing period after ‘prescribed’, and I think this statement needs a reference added.

Line 50 – why are young people especially vulnerable, and why specifically across college campuses? Presumably this may be related to increasing academic pressures, and because stimulant use can enhance concentration and therefore studying ability, but this is just my own interpretation and it isn’t actually clear. Clarification and a reference would strengthen.

Line 72 – define ‘toxic stress’ as not clear how this differs from other types of stress or if it is the same as chronic stress?

Line 76 – additional bracket before i.e.

General comment – while they are not stimulants, there is evidence that ACEs are a factor in non-medical benzodiazepine use and also opioid medication, as well as substances in general. So it might be useful in terms of rationale to include that there is evidence that ACEs predicts other types of drug use, including non-medical prescription use, for similar self-medicating reasons. In turn, this does suggest the potential for prescribed stimulants to be misused as well for similar reasons. A recent systematic review on ACEs and substance use that I came across may be the sort of paper that could be used as support: https://journals.sagepub.com/doi/10.1177/11782218231193914

General comment - Given the importance of understanding factors informing interventions for gender- and racially-diverse populations, this could be introduced in the introduction to say that, not only is the influence of ACEs uncertain, there is further uncertainty around whether intersecting characteristics such as gender identity and ethnicity may further pose issues for intervention development. Evidence highlights that people from minority groups face barriers to substance use support, so I think this issue needs introduced given it is an important finding/consideration from the findings.

Methods

Line 99 – NPSU is written here as nonmedical prescription stimulant misuse, and this is different from how it is written in lines 50 and 102 (and possibly others), so check for consistency with terms. Also if the acronym is used at line 50, the term doesn’t need fully written out again later (like in 102 and a few points later on so double check the full manuscript).

Line 104 – missing bracket after purposes.

Line 104 – Adverse Childhood Experiences is written out previously so the acronym could be used here.

Line 120 – Very clear inclusion/exclusion criteria which is helpful. Is there a justification as to why only studies in the last 10 years were included? I only say this because given there is such a small number of included studies, did the authors explore if increasing the date span might provide any further evidence?

Line 121 – I also have questions around why only English studies were included. Given 196 articles were excluded, although some of these were systematic reviews and meta-analyses, this still suggests quite a few non-english studies were excluded. Given the ability to translate articles easily using free online software, this is problematic to me. What was the reason behind this exclusion? If there is not a clear reason, this type of decision could be criticised for maintaining inequalities around the inclusion of Western-centric research over and above all other research. Further, this limitation is not discussed or included in the limitation section.

Line 142 – I see that the reader is directed to the PRISMA diagram here, but is there a reason the diagram itself is not here and is later on? That seems strange to me.

Results

Despite a small number of included studies, the number of participants across the studies is very high (over 46,500), and I think this is a notable strength in terms of how convincing the conclusions are.

Line 168 – the authors note differentiating between medications is the third aim, but it is listed as point ‘d’ in the aim at the end of the introduction section, so should it not be the fourth aim?

Line 179 – the authors discuss their desire to report demographic factors and so they include both sex and gender here, but then only report gender in the body of the paragraph. If the included studies report gender rather than sex, I think that ‘sex’ as a demographic could be removed since ‘gender’ is more accurate.

Line 188 – could a reference be provided to support the statement ‘in line with recent literature’.

Lines 188-190 – similar to my comment around gender vs sex wording, the authors use ‘race’ in the section heading, then ‘ethnicity’ in brackets at line 180, and then race again at lines 188-190. Consideration around which term is more accurate is necessary. This will be dependent on what the included studies report, but ethnicity and race are not interchangeable terms. I also note that they are used interchangeably at line 136 and in the table 1 heading and line 246 so perhaps double check the manuscript for other examples too.

Discussion

Line 250 – missing ‘s’ after ACE

Line 253 – I think the authors could add in about the prevalence of ACEs as well. Given how many young people experience ACEs (as briefly highlighted in the intro) then this would provide further rationale as to why they should be integrated into drug interventions moving forward if they are a factor in future NPSU.

General comment - Additionally, further consideration around what treatment/interventions might look like would be beneficial, for example a brief discussion around the need for more trauma-informed care within substance use treatment would strengthen this section and there is certainly a wealth of existing research that supports this, even if not specific to NPSU.  

General comment - I do think there needs to be some acknowledgement of the critiques around ACEs which are quite well established. For example, there are challenges cited in how ACEs are measured and challenges around definitions. This could mean additional complexity when exploring exactly what component of an ACE predicts future NPSU. Relatedly, consideration of how other known factors such as poverty, housing insecurity, community violence, bullying, etc. will likely have a mediating, if not direct effect too. While this is outside the scope of the paper, I think this consideration of a multitude of other factors (and which ACEs often overlook) needs at least mentioned.

General comment - Good to see consideration of intersecting vulnerabilities such as gender diverse young people and those from ethnic minority backgrounds. As well as discussing their increased risk in terms of ACEs and substance use, I think it is important to also note that gender diverse people and minority ethnic groups face barriers to drug treatment due issues like stigma, discrimination, and cultural barriers. Again this highlights the necessity of targeted interventions. This links to me earlier comment/suggestion re. the intro. 

Author Response

Introduction

  • Line 44 – missing period after ‘prescribed’, and I think this statement needs a reference added.
    • This change has been made.
  • Line 50 – why are young people especially vulnerable, and why specifically across college campuses? Presumably this may be related to increasing academic pressures, and because stimulant use can enhance concentration and therefore studying ability, but this is just my own interpretation and it isn’t actually clear. Clarification and a reference would strengthen.
    • Thank you for your comment. Clarification for NPSU across college campuses was provided, as well as references.
  • Line 72 – define ‘toxic stress’ as not clear how this differs from other types of stress or if it is the same as chronic stress?
    • This wording has been changed for clarity.
  • Line 76 – additional bracket before i.e.
    • Thank you for your attention to detail. This has been addressed.

General comment - Given the importance of understanding factors informing interventions for gender- and racially-diverse populations, this could be introduced in the introduction to say that, not only is the influence of ACEs uncertain, there is further uncertainty around whether intersecting characteristics such as gender identity and ethnicity may further pose issues for intervention development. Evidence highlights that people from minority groups face barriers to substance use support, so I think this issue needs introduced given it is an important finding/consideration from the findings.

 Methods

  • Line 120 – Very clear inclusion/exclusion criteria which is helpful. Is there a justification as to why only studies in the last 10 years were included? I only say this because given there is such a small number of included studies, did the authors explore if increasing the date span might provide any further evidence?
    • We appreciate your comment. Extending the date span did not result in any further studies being included.
  • Line 121 – I also have questions around why only English studies were included. Given 196 articles were excluded, although some of these were systematic reviews and meta-analyses, this still suggests quite a few non-english studies were excluded. Given the ability to translate articles easily using free online software, this is problematic to me. What was the reason behind this exclusion? If there is not a clear reason, this type of decision could be criticised for maintaining inequalities around the inclusion of Western-centric research over and above all other research. Further, this limitation is not discussed or included in the limitation section.
    • We greatly appreciate this suggestion. Under ideal circumstances, grant funding, and access to translation services, that approach would have been considered. We have included this as a limitation and a suggestion for future research in the discussion, which reads: Furthermore, the systematic review is not without limitations. The study did not include non-English articles, which might exclude relevant studies and/or result in language bias. Future research should consider including non-English articles if appropriate translation resources are available
  • Line 142 – I see that the reader is directed to the PRISMA diagram here, but is there a reason the diagram itself is not here and is later on? That seems strange to me.
    • Thank you for your comment. We will ask the publisher to move the diagram closer to its original mention.

Results

  • Line 168 – the authors note differentiating between medications is the third aim, but it is listed as point ‘d’ in the aim at the end of the introduction section, so should it not be the fourth aim?
    • Thank you for your comment. This has been addressed.
  • Line 179 – the authors discuss their desire to report demographic factors and so they include both sex and gender here, but then only report gender in the body of the paragraph. If the included studies report gender rather than sex, I think that ‘sex’ as a demographic could be removed since ‘gender’ is more accurate.
    • Thank you. We agree with your comment and have removed ‘sex’ from our operationalization of demographic factors.
    •  
  • Line 188 – could a reference be provided to support the statement ‘in line with recent literature’.
    • Thank you. A reference was added.
  • Lines 188-190 – similar to my comment around gender vs sex wording, the authors use ‘race’ in the section heading, then ‘ethnicity’ in brackets at line 180, and then race again at lines 188-190. Consideration around which term is more accurate is necessary. This will be dependent on what the included studies report, but ethnicity and race are not interchangeable terms. I also note that they are used interchangeably at line 136 and in the table 1 heading and line 246 so perhaps double-check the manuscript for other examples too.
    • Thank you for this great suggestion. Race and ethnicity were clarified throughout the paper as separate terms.

Discussion

  • Line 253 – I think the authors could add in about the prevalence of ACEs as well. Given how many young people experience ACEs (as briefly highlighted in the intro) then this would provide further rationale as to why they should be integrated into drug interventions moving forward if they are a factor in future NPSU.
    • We appreciate this comment and added in a sentence about the prevalence of ACEs.
  • General comment - Additionally, further consideration around what treatment/interventions might look like would be beneficial, for example, a brief discussion around the need for more trauma-informed care within substance use treatment would strengthen this section and there is certainly a wealth of existing research that supports this, even if not specific to NPSU.  
    • We greatly appreciate this comment, and we included a brief discussion around the need for trauma-informed care across healthcare settings (specifically substance use settings).
  • General comment - I do think there needs to be some acknowledgment of the critiques around ACEs which are quite well established. For example, there are challenges cited in how ACEs are measured and challenges around definitions. This could mean additional complexity when exploring exactly what component of an ACE predicts future NPSU. Relatedly, consideration of how other known factors such as poverty, housing insecurity, community violence, bullying, etc. will likely have a mediating, if not direct effect too. While this is outside the scope of the paper, I think this consideration of a multitude of other factors (and which ACEs often overlook) needs at least mentioned.
    • We agree with this comment. We included an acknowledgment of this in our discussion section.
  • General comment - Good to see consideration of intersecting vulnerabilities such as gender diverse young people and those from ethnic minority backgrounds. As well as discussing their increased risk in terms of ACEs and substance use, I think it is important to also note that gender diverse people and minority ethnic groups face barriers to drug treatment due issues like stigma, discrimination, and cultural barriers. Again this highlights the necessity of targeted interventions. This links to me earlier comment/suggestion re. the intro. 
  • Thank you for this suggestion. We added a discussion of barriers experienced by drug treatment.

Round 2

Reviewer 2 Report

Comments and Suggestions for Authors

Thank you for addressing our questions/comments.